# LLC DC-DC Converter Performances Improvement for Bidirectional Electric Vehicle Charger Application

Houssein Al Attar [1,*], Mohamed Assaad Hamida [1], Malek Ghanes [1] and Miassa Taleb [2]

1   Ecole Centrale de Nantes, LS2N, UMR CNRS 6004, 1 Rue de la Noe, BP 92101, CEDEX 3, 44321 Nantes, France; mohamed.hamida@ec-nantes.fr (M.A.H.); malek.ghanes@ec-nantes.fr (M.G.)
2   Renault Group, Technocentre, 1 Avenue du Golf, 78280 Guyancourt, France; miassa.taleb@renault.com
*   Correspondence: houssein.al-attar@ec-nantes.fr

**Abstract:** Electric Vehicle (EV) bidirectional charger technology is growing in importance. It defines the fact of returning the electricity stored in the batteries of EV to Grid (V2G), to Home (V2H), to Load (V2L), or in one word V2X mode. The EV onboard charger is divided into two parts: AC-DC and DC-DC converters. The isolated bidirectional DC-DC LLC resonant converter is used to improve the charger efficiency within both battery power and voltage ranges. It is controlled by varying the switching frequency based on a small signal modeling approach using the gain transfer function inversion method. The dimensions of the DC-DC LLC converter directly affect the charger cost. Moreover, they cause an important control frequency saturation zone, especially in V2X mode, where the switching frequency is out of the feasibility zone. The new challenge in this paper is to design an optimization strategy to minimize the LLC converter cost and improve the control frequency feasibility zone, for a wide variation of battery voltage and converter power, in the charging (G2V) and discharging (V2X) modes simultaneously. For our best knowledge, this optimization problem, in the case of a bidirectional (G2V and V2X) charger, is not yet considered in the literature. An optimal design that considers the control stability equations in the optimization algorithm is elaborated. The obtained results show a significant converter cost decrease and important expansion of control frequency feasibility zones. A comparative study between initial and optimized values, in G2V and V2X modes, is generated according to the converter efficiency.

**Keywords:** electric vehicle charger; DC-DC LLC converter; optimization strategy; control frequency feasibility; G2V mode; V2X mode

## 1. Introduction

### 1.1. Overview

Energy transport is today a major component of the energy transition. By using EV as a transport vector, it becomes a significant technology allowing different uses. G2V defines the concept that the EV batteries are charged from the power grid. The energy stored in the EV battery can then be used as a current source (V2G), or a voltage source (V2L/V2H). An EV in V2X mode offers reactive and active power regulation, load balancing, and tracking of variable renewable energy sources. The battery charger must then be able to ensure the conversion in both directions of energy flow and thus becomes bidirectional: charging/discharging. There are two types of EV chargers: onboard or AC chargers that can be integrated into the vehicle, and offboard or DC chargers that cannot. EV DC charging standards are mostly used for fast charging due to their higher power capabilities (>50 kW) and output voltages [1]. The onboard charger is often designed for lower kilowatts of power transfer and adds a significant weight to the vehicle. It is responsible for the final stage of the battery pack charging inside the EV. An equivalent scheme for a bidirectional battery charger system is presented in Figure 1.

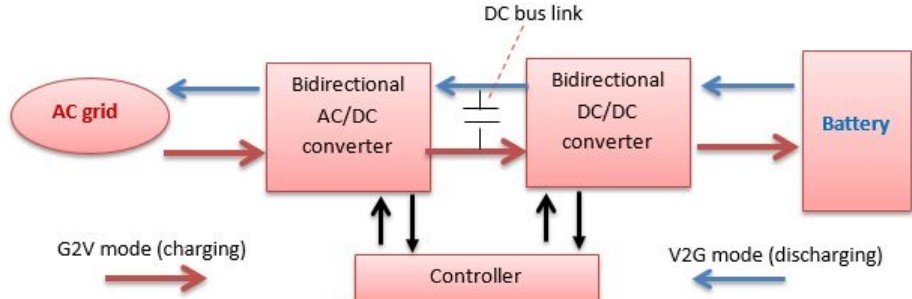

**Figure 1.** EV bidirectional charger topology.

The two modes G2V and V2X are provided by the bidirectional converters: AC-DC and DC-DC converters. Isolated bidirectional DC-DC converters [2–4] are used as a key device for interfacing the storage devices between a DC bus and the battery with high power applications. There are many topologies of isolated converters, such as Dual Active Bridge (DAB) converters [5] and Series Resonant Converters (SRC) [6], but they suffer from soft switching property loss at high input voltage and with a light load. DC-DC LLC resonant converters [7–10] are commonly selected for applications that demand a high power density, such as the EV charger.

An onboard battery charger is constrained by sizing, weight, and cost. It should be also implemented either with unidirectional or bidirectional power flow (Figure 1). The dimensions of the LLC converter are an essential part of the onboard charger sizing. The advantages of decreasing DC-DC LLC converter dimensions include reduced sizing, weight, and cost of the charger and improved reliability.

For DC-DC LLC resonant converters, the most frequently adopted modulation strategy is Pulse Frequency Modulation (PFM). The switching frequency feasible zone is defined between a minimum and maximum authorized value to guarantee Zero Voltage Switching (ZVS) condition. This frequency feasibility condition provides a cost minimization challenge related to both software (FPGA operation) and hardware (charger sizing) implementations in the EV charger. However, when the PFM strategy is adopted for wide input/output range application in the onboard battery charger in V2X mode, a wide operating switching frequency range is required to meet the system voltage gain requirement. A wide switching frequency range causes soft switching operation loss, which results in low conversion efficiency and control performances.

*1.2. State of the Art*

Several research studies have been elaborated to optimally design the DC-DC LLC resonant converter. In [11], an optimal design methodology for the LLC resonant converter in battery charging applications is studied based on Time-Weighted Average Efficiency (TWAE). The TWAE index presents the average weight of conversion efficiency, which covers power losses of major components, during the battery charging period and it is proposed as the objective function to optimize the converter parameters. The optimization variables are the three resonant parameters: $L_r$, $C_r$, and $L_m$ presented in Section 2. Each of them is bound by physical constraints. The resonant frequency is often set between the minimum and maximum frequency to achieve ZVS. The optimization algorithm is based on calculating the current, voltage, and frequency of the converter under typical charging conditions. An important TWAE is achieved. Moreover, instead of minimizing the components stresses, a proper choice of the converter parameters resulting in an improved MOSFETS control operation has been performed in [12]. The power MOSFETS drain-sources have been derived to realize the ZVS operation. Soft switching is achieved for all power devices under all operating conditions. The optimized design is achieved for different regulated output voltages (35–165 V) under different loads (0–3 A) and input voltages (320–370 V).

In [3], a power converter with a multi-operating mode is proposed. The topology can work as a bidirectional DC-DC converter. The controller is implemented based on a state-space averaging model. A weight decrease in the power converter in the electric vehicle is obtained. A double-boost DC/DC converter for electric vehicles is designed in [13]. A feedforward double closed-loop control is proposed based on the small-signal model. The proposed converter showed an improved efficiency compared to other topologies.

In [14], an optimal algorithm is applied to a three-port multidirectional DC-DC converter to increase efficiency and reduce energy costs using a droop control method. A novel multilevel DC-DC converter is proposed in [15] to enhance the performances of traditional multilevel DC/DC converters using an impedance source that reduces the number of sub-modules and improves the system efficiency. Ref. [16] presents an adjustable robust optimization to improve the performances of the microgrid system by increasing the absorption ratio of renewable energy in G2V and V2G modes of EV.

In [17], an optimization methodology is designed to obtain optimal scheduling of smart charging of EVs considering the requirements, such as time specifications as is the case. An optimal design of a multi-resonant converter for interfacing electrolyser stack to DC voltage source is studied in [18]. The design improves ZVS but not throughout the whole operating range. Isolated single input, dual output DC-DC converters for EVs is studied in [19]. The proposed control strategy consists of regulating two different output voltages by varying the duty cycle and switching frequency to improve the converter efficiency.

Furthermore, Refs. [8,20] show an optimal design methodology under the worst operation condition (minimum input voltage with maximum output power or maximum input voltage with minimum output power). Some constraints are included to obtain the suitable design area, such as operation mode, voltage stress for resonant capacitor, ZVS operation for primary switches, and resonant tank root-mean-square current. The main drawback related to this methodology is based on the fact that the LLC converter is designed to operate in the above resonant frequency region, thus the control instability issue under light load operation may occur due to the circuit parasitics.

Refs. [21–23] highlight an optimized transformer design for LLC converter to reduce core volume and conduction losses in the transformer windings for high power density. In addition to that, it provides the necessary isolation and required voltage-conversion and magnetizing inductance for efficient converter operation. It also makes a significant contribution to the weight and size of the overall converter.

In [24], computer-aided design optimization is proposed to design LLC converters to optimize the converter efficiency. A mode solver technique is proposed to handle LLC converter steady-state solutions. The mode solver utilizes numerical non-linear programming techniques to solve LLC-state equations and determine operation mode. The objective function is the calculated efficiency. The fmincon function of MATLAB optimization toolbox is applied as the optimizer to solve this non-linear, constrained, and continuous optimization problem. In [25,26], an optimized design of parameters of the main circuit of the LLC resonant converter is proposed to reach the maximum controllability characteristics. A selection of compromise between conduction losses, switching losses and regulation range with stable ZVS condition is considered. The optimization model is developed based on the First Harmonic Approximation (FHA) analysis.

However, the proposed optimization strategies in the literature are mostly intended to minimize the cost and weight of the DC-DC LLC converter and guarantee soft-switching property in one operation mode, i.e., G2V mode [11,12,20,24–26]. Moreover, the LLC converter is designed based on a reduced battery voltage (12–48 V) [8,20–26] and converter power (20–1000 W) [8,12,20–26]. For our best knowledge, no optimization algorithm, which takes into consideration the DC-DC LLC converter behavior in the two operation modes G2V and V2X for a bidirectional charger application, is presented to minimize the cost and increase the soft-switching range for a wide variation of the battery voltage and power.

### 1.3. Contributions

With respect to the literature, the main contribution of this article is to design an optimization strategy to minimize the DC-DC LLC converter cost and to improve the control performances and the EV charger efficiency by increasing the soft switching operating zone, in both G2V and V2X modes, for a wide variation of the battery voltage (240–430 V) and converter power (0–11,000 W) . The control constraints are based on the control strategy studied in our conference paper [27], where PFM based on a gain inversion method is presented to regulate DC bus voltage around a certain setpoint (450 V).

The contributions of this paper can be summarized as follows:

- Optimal design of LLC converter dimensions;
- EV charger cost and sizing minimization;
- Important improvement of the control frequency feasibility zone for V2X mode;
- Total reduction in the control frequency saturation zone for G2V mode;
- Control performances and converter efficiency improvement for EV bidirectional charger application with wide battery voltage and power variation.

The paper is organized as follows: Section 2 provides the problem statement. In Section 3, the proposed optimization strategy is presented. In Section 4, the simulation results are highlighted. In Section 5, a general conclusion is drawn.

## 2. Problem Statement

The bidirectional DC-DC LLC resonant converter is presented in Figure 2. The resonant tank consists of a series capacitor $C_r$, a series inductor $L_r$, a parallel magnetizing inductor $L_m$ and a transformer with a turns-ratio of $n$ where $C_1$ is the DC bus capacitor, $V_{DC}$ is the DC bus voltage, $V_{bat}$ is the battery voltage and we will design by $P$ the converter power. The resonant tank is directly related to a high frequency transformer that allows galvanic isolation of the charger.

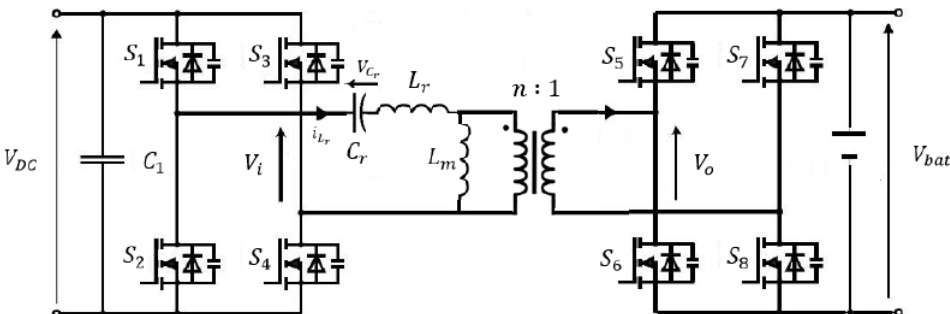

**Figure 2.** DC-DC LLC converter topology.

It is essential to define a representative dynamic model of the DC-DC LLC resonant converter in order to implement a control strategy for this converter. The large-signal models based on Extending Describing Function (EDF) [28–30] have been proposed for the LLC converter. These models provide enough dynamic information of the LLC converter when large-signal transient disturbance occurs. However, these models are still non-linear making the design of the control more complex. Many classical control laws, such as sliding mode control and PID, are proposed in the literature based on large or small-signal models [30–33]. The proposed strategies for this converter have a battery voltage regulation (if this is not imposed) or a DC current regulation, where the battery voltage varies over a reduced range (12–60 V).

DC bus voltage regulation based on a gain inversion method for DC-DC LLC with wide variations in both the battery voltage (240–430 V) and power (0–11 kW) is envisaged. The h-bridge of the LLC converter ensures a constant DC voltage at the DC bus by MOSFET signals regulation. The dynamics of the LLC converter are investigated using

the small-signal modeling technique based on First Harmonic Approximation (FHA) methodology [34].

PFM strategy consists of varying the switching frequency of MOSFET control signals. For G2V mode, the power MOSFETs of the full-bridge in the primary side (respectively, in the secondary side for V2X mode) of the transformer are controlled in complementary at duty 0.5 ignoring the dead-time, whereas the diodes of the full -bridge in the secondary side (respectively, in the primary side for V2X mode) are used for rectification.

### 2.1. G2V Mode

FHA is based on the following assumptions:

- The input voltage is presented as an ideal sinusoidal voltage source, which represents only the fundamental component ignoring all the higher-order harmonics;
- The output filter capacitor and the primary side leakage inductance of the transformer are ignored.

The input voltage waveform of the resonant tank in Figure 2 can be expressed in Fourier series, whose fundamental component $V_{if}$ has the following expression (1) [34]:

$$V_{if} = \frac{2\sqrt{2}}{\pi} \cdot V_{DC} \tag{1}$$

For the secondary side, the rectifier is driven by a square wave output voltage with a fundamental component $V_{of}$ (2) [34]:

$$V_{of} = \frac{2\sqrt{2}}{\pi} \cdot n \cdot V_{bat} \tag{2}$$

The equivalent model of the LLC resonant converter (Figure 2), using small-signal modeling with FHA, can be obtained as shown in Figure 3 [34,35].

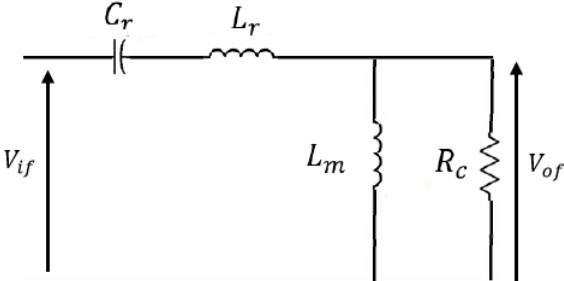

**Figure 3.** FHA equivalent model of LLC converter in G2V mode.

The equivalent ac resistance $R_c$ is given, based on FHA [34], by (3):

$$R_c = n^2 \frac{8}{\pi^2} \frac{V_{bat}^2}{P} \tag{3}$$

Based on the equivalent model of the LLC converter in G2V mode (Figure 3), the gain transfer function magnitude is given by (4):

$$|G| = \frac{V_{of}}{V_{if}} = \frac{n.V_{bat}}{V_{DC}} = \frac{|-R_c \cdot L_m \cdot C_r \cdot w^2|}{|-L_m \cdot L_r \cdot C_r \cdot w^3 \cdot j - (R_c \cdot C_r \cdot (L_m + L_r)) \cdot w^2 + L_m \cdot w \cdot j + R_c|} \tag{4}$$

with $w = 2\pi f$, where $f$ is the switching frequency.

The DC-DC LLC transfer function is represented by the DC bus voltage/battery voltage ratio. A switching frequency control of the H-bridge will adapt the gain of the DC-DC and allow us to regulate the DC bus voltage to a certain setpoint (450 V).

A PFM control strategy based on a gain inversion method is developed to ensure a more stable response with respect to DC current disturbances at the DC-DC input. The

gain inversion makes it possible to obtain an expression (feedforward) of the switching frequency which depends on the parameters of the resonant circuit, the demanded power, the DC bus voltage and the battery voltage. The feedforward switching frequency $f_{0c}$ is defined based on (4), to ensure a DC bus voltage at the input of the LLC converter, as a solution of the following Equation (5):

$$a \cdot w^6 + b \cdot w^4 + c \cdot w^2 + d = 0 \tag{5}$$

where the parameters $a$, $b$, $c$ and $d$ are expressed as follows:

$$a = L_m{}^2 \cdot L_r{}^2 \cdot C_r{}^2 \tag{6}$$

$$b = R_c^2 \cdot C_r^2 \cdot (L_r + L_m)^2 - 2 \cdot L_m^2 \cdot L_r \cdot C_r - R_c^2 \cdot L_m^2 \cdot C_r^2 \cdot \frac{V_{DC}^2}{(n \cdot V_{bat})^2} \tag{7}$$

$$c = L_m^2 - 2 \cdot R_c^2 \cdot C_r \cdot (L_m + L_r) \tag{8}$$

$$d = R_c^2 \tag{9}$$

By taking $y = w^2$, (5) can be transformed into the following cubic Equation (10):

$$a \cdot y^3 + b \cdot y^2 + c \cdot y + d = 0 \tag{10}$$

By solving (10) using what is known as the Tschirnhaus–Vieta approach for cubic equations, the feedforward switching frequency $f_{0c}$ can be obtained as follows:

$$p = \frac{3 \cdot a \cdot c - b^2}{3 \cdot a^2} \tag{11}$$

$$q = \frac{2 \cdot b^3 - 9 \cdot a \cdot b \cdot c + 27 \cdot a^2 \cdot d}{27 \cdot a^3} \tag{12}$$

$$M = 2\sqrt{\frac{-p}{3}} \tag{13}$$

$$\phi = arccos(\frac{3 \cdot q}{M \cdot p}) \tag{14}$$

$$f_{0c} = \frac{1}{2\pi}(\sqrt{M \cdot cos(\frac{\phi}{3}) - \frac{b}{3 \cdot a}}) \tag{15}$$

In Figure 4, $f_{0c}$ is presented with respect to the battery voltage and power variations.

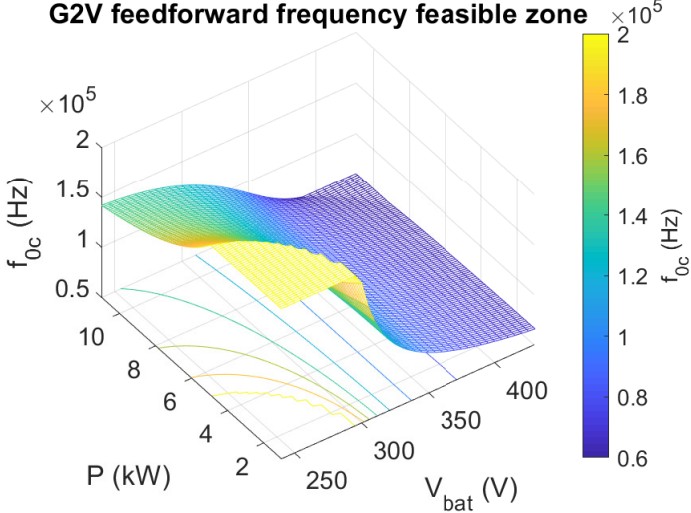

**Figure 4.** Feedforward frequency in G2V mode with respect to battery voltage and power variation.

According to Figure 4, the yellow zone represents an operating zone where the control frequency is saturated at $f_{max}$ (maximal authorized switching frequency). There is a small frequency saturation zone in G2V mode that affects the ZVS property for the LLC converter. This saturation zone provides a low efficiency and causes an important DC bus voltage regulation error with PFM strategy.

### 2.2. V2X Mode

In V2X mode, the input voltage waveform of the resonant tank, in Figure 2, can be expressed as in (16) [27]:

$$V_{if} = \frac{2\sqrt{2}}{\pi} \cdot n \cdot V_{bat} \tag{16}$$

The rectifier is driven by a square wave output voltage with a fundamental component $V_{of}$ (17):

$$V_{of} = \frac{2\sqrt{2}}{\pi} \cdot V_{DC} \tag{17}$$

The equivalent model of the LLC resonant converter (Figure 2) can be obtained, using small-signal modeling with FHA, as shown in Figure 5 [27].

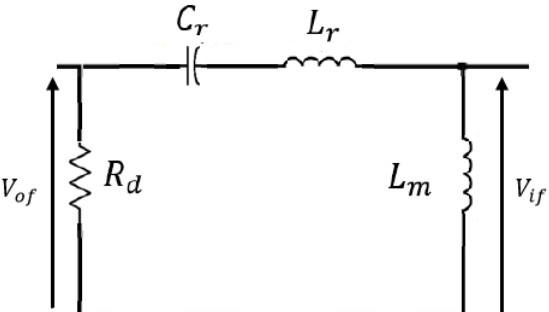

**Figure 5.** FHA equivalent model of LLC converter in V2X mode.

The equivalent ac resistance $R_d$ is given, based on FHA [27], by (18):

$$R_d = \frac{8}{\pi^2} \frac{V_{DC}^2}{P} \tag{18}$$

Based on the FHA equivalent model of the LLC converter in V2X mode in Figure 5, the gain transfer function can be expressed as in (19):

$$|G| = \frac{V_{of}}{V_{if}} = \frac{V_{DC}}{n \cdot V_{bat}} = \frac{|R_d . C_r \cdot wj|}{|1 - L_r \cdot C_r \cdot w^2 + R_d \cdot C_r \cdot wj|} \tag{19}$$

With $w = 2\pi f$, where $f$ is the switching frequency.

The feedforward switching frequency $f_{0d}$, resulting from the gain inversion method, is defined based on (19) as a solution of the following Equation (20):

$$A \cdot w^4 + B \cdot w^2 + 1 = 0 \tag{20}$$

where the parameters $A$ and $B$ are defined in (21) and (22), respectively:

$$A = L_r^2 \cdot C_r^2 \tag{21}$$

$$B = R_d^2 \cdot C_r^2 - 2 \cdot L_r \cdot C_r - \frac{(n \cdot V_{bat})^2}{V_{DC}^2} \cdot R_d^2 \cdot C_r^2 \tag{22}$$

By taking $v = w^2$, (20) can be transformed into the following Equation (23):

$$A \cdot v^2 + B \cdot v + 1 = 0 \tag{23}$$

By solving (23), the feedforward switching frequency $f_{0d}$ can be expressed as in (24):

$$f_{0d} = \frac{1}{2\pi}\sqrt{\frac{-B + \sqrt{B^2 - 4 \cdot A}}{2 \cdot A}} \tag{24}$$

In Figure 6, $f_{0d}$ is presented with respect to the battery voltage and power variations.

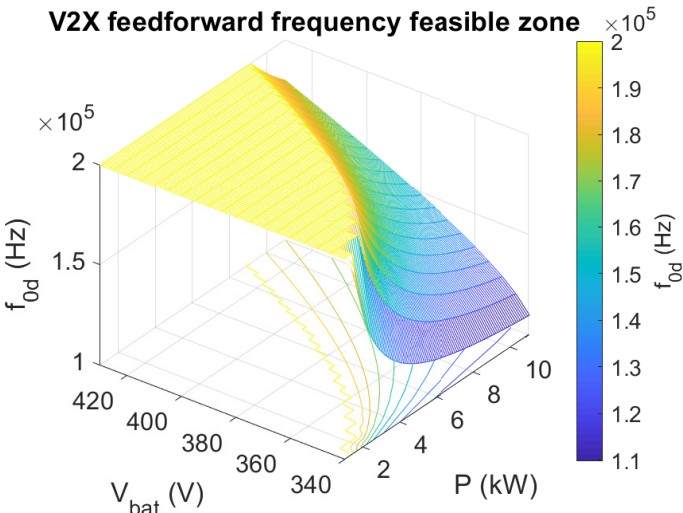

**Figure 6.** Feedforward frequency in V2X mode with respect to battery voltage and power variation.

Based on Figure 6, there is an important frequency saturation zone in V2X mode (much higher than in G2V mode), giving a low efficiency and low control performances with PFM strategy in the reversible operation mode of the EV charger.

In summary, the DC-DC LLC converter parameters, such as $C_r$, $L_r$, and $L_m$, directly affect the onboard charger performances. On one hand, they have an important effect on the control frequency feasibility for a wide operating zone. The feedforward switching frequency, in G2V and V2X modes, depends essentially on them. On the other hand, they affect the onboard charger sizing and cost.

## 3. Proposed Optimization Strategy Design

The aim is to design an optimization methodology to minimize the DC-DC LLC converter cost and to improve the control performances by taking into consideration the switching frequency feasibility equations with respect to the wide operating zone in both G2V and V2X modes.

### 3.1. LLC Parameters Effect
### 3.1.1. G2V Mode

The equivalent LLC model based on small-signal modeling in Figure 3 shows that $L_r$, $C_r$, and $L_m$ are the main parameters to define the resonant circuit sizing. Moreover, the LLC gain transfer function presented in (4) depends on these parameters. Thus, any change in the value of any of them will affect the feedforward switching frequency $f_{0c}$ (15). The effect of the variation of $L_m$, $L_r$, and $C_r$ in $f_{0c}$ is studied for G2V mode for some operating points with fixed DC bus voltage and transformer ratio.

It should be noted that $L_{m0}$, $L_{r0}$, and $C_{r0}$ are the initial values of the DC-DC LLC converter.

- $L_r$ effect:
  With $L_m = L_{m0}$, $C_r = C_{r0}$, $V_{bat} = 420$ V and by varying $P$ ($P_{min} < P < P_{max}$), $f_{0c}$ is presented in Figure 7 for different operating points with respect to the variation of $L_r$.

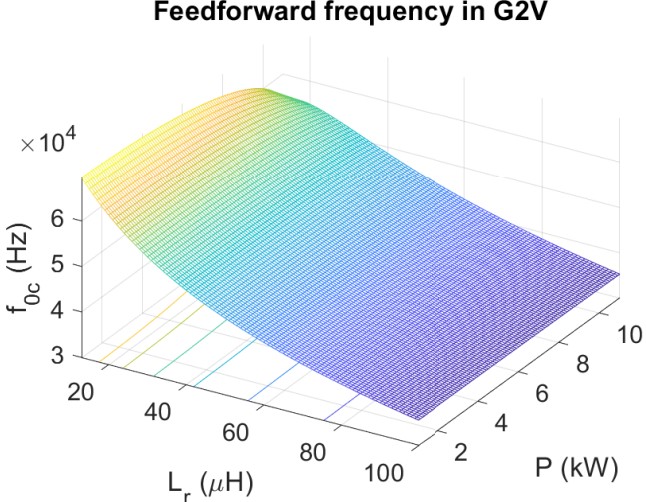

**Figure 7.** G2V feedforward frequency in function of $L_r$.

It is clearly seen that $f_{0c}$ increases with the decrease in $L_r$.

- $C_r$ effect:
  With $L_m = L_{m0}$, $L_r = L_{r0}$, $V_{bat} = 420$ V and by varying $P$, $f_{0c}$ is presented in Figure 8 for different operating points with respect to the variation of $C_r$.

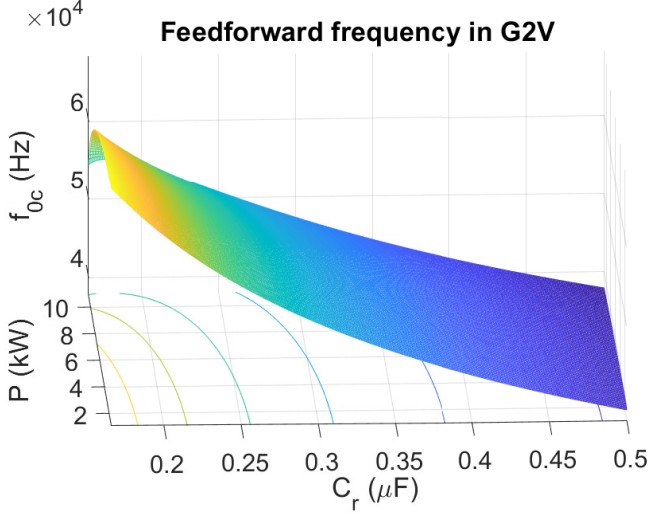

**Figure 8.** G2V feedforward frequency in function of $C_r$.

It is confirmed that $f_{0c}$ increases with the decrease in $C_r$ too, but at a lower rate than in the case of $L_r$.

- $L_m$ effect:
  With $L_r = L_{r0}$, $C_r = C_{r0}$, $V_{bat} = 420$ V and by varying $P$, $f_{0c}$ is presented in Figure 9 for different operating points with respect to the variation of $L_m$.

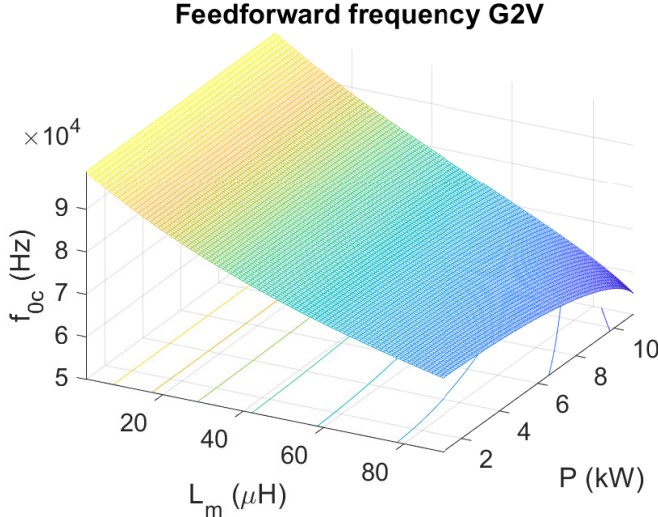

**Figure 9.** G2V feedforward frequency in function of $L_m$.

It is clearly seen that $f_{0c}$ increases with the decrease in $L_m$.

### 3.1.2. V2X Mode

In V2X mode, the equivalent LLC model, presented in Figure 5, is based on the small-signal modeling with FHA. Thus, it should be noted that $L_m$ does not affect the feedforward switching frequency $f_{0d}$ (24) obtained from (19). The effect of the variation of $L_r$ and $C_r$ in $f_{0d}$ is studied for V2X mode for some operating points with fixed DC bus voltage and transformer ratio.

- $L_r$ effect:
  With $C_r = C_{r0}$, $V_{bat} = 420$ V and by varying $P$, $f_{0d}$ is presented in Figure 10 for different operating points with respect to the variation of $L_r$.

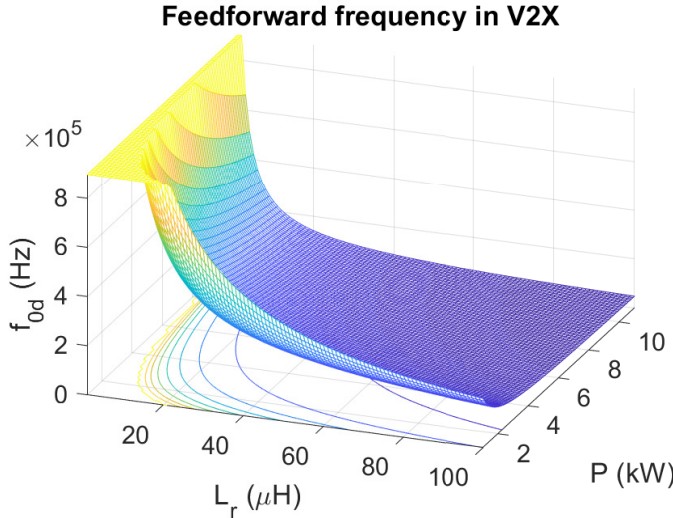

**Figure 10.** V2X feedforward frequency in function of $L_r$.

It is clearly seen that $f_{0d}$ increases with the decrease in $L_r$.

- $C_r$ effect:
  With $L_r = L_{r0}$, $V_{bat} = 420$ V and by varying $P$, $f_{0d}$ is presented in Figure 11 for different operating points with respect to the variation of $C_r$.

**Feedforward frequency in V2X**

**Figure 11.** V2X feedforward frequency in function of $C_r$.

It is confirmed that $f_{0d}$ increases with the decrease in $C_r$ too.

### 3.1.3. Summary

Table 1 summarizes the effect of the decrease in the LLC parameters, $L_r$, $C_r$, and $L_m$, on the feedforward frequencies $f_{0c}$ and $f_{0d}$. Based on this study of the LLC parameters effect on the feedforward switching frequency in G2V and V2X modes, it is confirmed that $f_{0c}$ and $f_{0d}$ are inversely proportional to $L_r$, $C_r$, and $L_m$. $f_{0c}$ and $f_{0d}$ should respect ZVS condition and not exceed the minimum and maximum authorized frequencies. Thus, there is a balance between cost and sizing minimization and control performances improvement. In the next section, an optimal algorithm design will be a new challenge to reach the target points.

**Table 1.** Effect of LLC parameters decrease on feedforward switching frequencies.

| LLC Parameters | $f_{0c}$ | $f_{0d}$ |
|:---:|:---:|:---:|
| $L_r$ | Increase | High increase |
| $C_r$ | Increase | Very low increase |
| $L_m$ | Increase | None |

### 3.2. Optimization Strategy Design

The aim of the optimization is to find the optimum $L_r$, $C_r$, and $L_m$ that minimize the cost and sizing of the charger and improve the control frequency feasibility in G2V and V2X modes simultaneously.

The optimization problem can be expressed by 5 decision variables as presented in (25):

$$X = \begin{bmatrix} L_r & C_r & L_m & f_{0c} & f_{0d} \end{bmatrix} = \begin{bmatrix} X(1) & X(2) & X(3) & X(4) & X(5) \end{bmatrix} \quad (25)$$

$f_{0c}$ is the feedforward switching frequency in G2V mode defined in (15).
$f_{0d}$ is the feedforward switching frequency in V2X mode defined in (24).

To obtain an optimization problem with respect to the battery voltage and power variation, the decision vector becomes:

$$X_{ij} = \begin{bmatrix} X(1)_{ij} & X(2)_{ij} & X(3)_{ij} & X(4)_{ij} & X(5)_{ij} \end{bmatrix} \quad (26)$$

where the indexes $i$ and $j$ represent the battery voltage and the converter power vectors, respectively, with:

$$V_{min} \leq V_{bat}(i) \leq V_{max}$$

$$P_{min} \leq P(j) \leq P_{max}$$

The optimal design of the resonant circuit should guarantee control stability in both G2V and V2X modes. In other words, the optimal values of $L_r$, $C_r$, and $L_m$ should improve the feedforward switching frequency zone, based on PFM strategy with the gain inversion method, to reduce the saturation zones presented in Figures 4 and 6. Control stability constraints based on the gain inversion should be added to the optimization problem. The feedforward switching frequencies $f_{0c}$ and $f_{0d}$, calculated using the optimal LLC parameters resulting from the optimization algorithm, are the solutions of (5) and (20) in G2V and V2X modes, respectively. Therefore, the optimization strategy should not only minimize the DC-DC LLC converter sizing and cost by minimizing the LLC parameters $L_r$, $C_r$, and $L_m$ but also generate an improved feedforward switching frequency zones in G2V and V2X modes according to the battery voltage and converter power variation.

To obtain the optimization target points, the objective function can be formulated as expressed in (27):

$$F(X) = \alpha_1 \cdot X(1) + \alpha_2 \cdot X(2) + \alpha_3 \cdot X(3) + C_{G2V}(X) + C_{V2X}(X) \tag{27}$$

where $\alpha_1$ and $\alpha_3$ represent the costs euro/H of the resonant and magnetizing inductors $L_r$ and $L_m$, respectively, and $\alpha_2$ represent s the cost euro/F of the resonant capacitor $C_r$.

$C_{G2V}(X)$ and $C_{V2X}(X)$ are the equations that define the control stability in G2V and V2X modes, respectively. They are based on (5) (for G2V mode) and (20) (for V2X mode) that allow us to obtain the feedforward switching frequencies for the whole operating zone. They can be expressed, in (28) and (29), respectively, as follows:

$$C_{G2V}(X) = a(X) \cdot (2\pi \cdot X(4))^6 + b(X) \cdot (2\pi \cdot X(4))^4 + c(X) \cdot (2\pi \cdot X(4))^2 + d \tag{28}$$

where $a$, $b$, $c$, and $d$ are defined in (6)–(9), respectively, in function of the LLC parameters $X(1)$ (or $L_r$), $X(2)$ (or $C_r$) and $X(3)$ (or $L_m$).

$$C_{V2X}(X) = A(X) \cdot (2\pi \cdot X(5))^4 + B(X) \cdot (2\pi \cdot X(5))^2 + 1 \tag{29}$$

where $A$ and $B$ are defined in (21) and (22), respectively, in function of $X(1)$ and $X(2)$.

The optimization problem includes linear and non-linear constraints resulting from hardware conditions and control requirements. The constraints of the variation of $L_r$, $C_r$ and $L_m$, related to the hardware design limitations, are expressed in (30)–(32):

$$\frac{L_{r0}}{10} \leq X(1) \leq L_{r0} \tag{30}$$

$$\frac{C_{r0}}{10} \leq X(2) \leq C_{r0} \tag{31}$$

$$\frac{L_{m0}}{10} \leq X(3) \leq L_{m0} \tag{32}$$

To guarantee ZVS property, the switching frequency, in each of G2V and V2X modes, should respect the following constraints formulated in (33) and (34), respectively:

$$f_{min} \leq X(4) \leq f_{max} \tag{33}$$

$$f_{min} \leq X(5) \leq f_{max} \tag{34}$$

where $f_{min}$ and $f_{max}$ are the minimum and maximum authorized switching frequencies, respectively.

There are also some constraints that should be defined, to obtain more accurate feedforward switching frequencies, i.e., solutions of the control stability equations, as follows in (35) and (36):

$$a(X) \cdot (2\pi \cdot X(4))^6 + b(X) \cdot (2\pi \cdot X(4))^4 + c(X) \cdot (2\pi \cdot X(4))^2 + d \geq 0 \tag{35}$$

$$A(X) \cdot (2\pi \cdot X(5))^4 + B(X) \cdot (2\pi \cdot X(5))^2 + 1 \geq 0 \tag{36}$$

As a result of this optimization algorithm, the global minimum vector $X_{ij}$ (26) for each operating point (battery voltage $V_{bat}(i)$ and converter power $P(j)$) can be obtained. To obtain the optimal values of the LLC converter parameters for the whole operating zone, the mean value of the global minimums of each LLC parameter is calculated as follows:

$$L_{r1} = \frac{\sum_{i=1}^{p} \sum_{j=1}^{q} X(1)_{ij}}{p \times q} \tag{37}$$

$$C_{r1} = \frac{\sum_{i=1}^{p} \sum_{j=1}^{q} X(2)_{ij}}{p \times q} \tag{38}$$

$$L_{m1} = \frac{\sum_{i=1}^{p} \sum_{j=1}^{q} X(3)_{ij}}{p \times q} \tag{39}$$

$L_{r1}$, $C_{r1}$, and $L_{m1}$ are the mean values of global minimums of $L_r$, $C_r$, and $L_m$, respectively. Based on the mean values $L_{r1}$, $C_{r1}$ and $L_{m1}$, the feedforward switching frequencies $f_{0c}$ and $f_{0d}$ can be obtained using (15) and (24) for the whole operating zone (battery voltage and converter power variation). The optimization algorithm can be summarized using the flow chart presented in Figure 12.

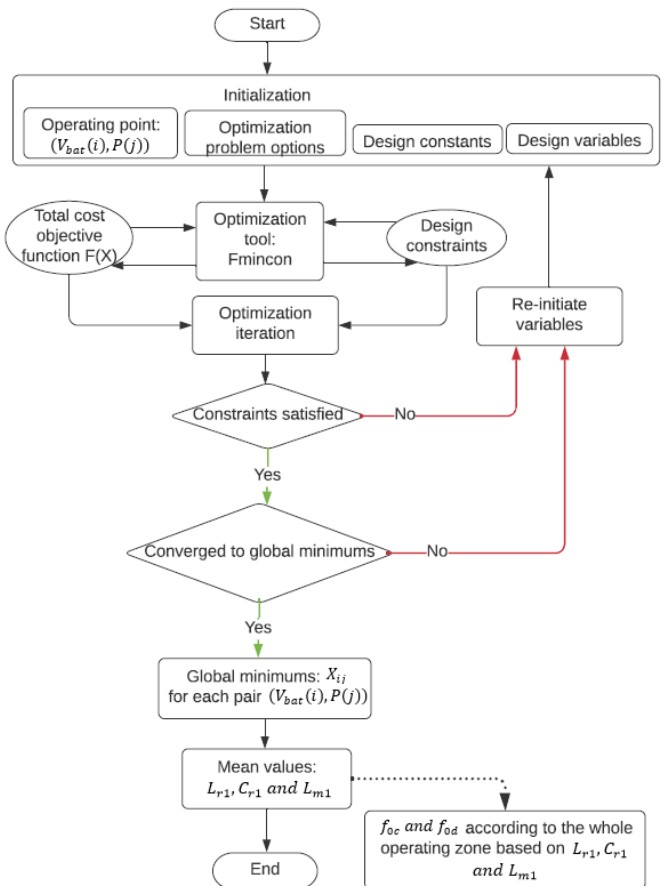

**Figure 12.** Optimization flow chart.

## 4. Simulations and Results

The optimization problem is solved using MATLAB optimization solver fmincon with an Intel Core i7 CPU @ 2.11GH. The DC-DC LLC converter and the control system are implemented in MATLAB/SIMULINK. It should be noted that the simulation parameters are those of a real model of DC-DC LLC converter used in electric vehicle charger. The simulations have been completed using Fixed Step Discrete solver (Sample time = 7.14286 $\times 10^{-8}$ s). The parameter settings are shown in Table 2.

**Table 2.** Settings table.

| $V_{min}$ | 240 V | $V_{max}$ | 430 V |
|---|---|---|---|
| $P_{min}$ | 1 kW | $P_{max}$ | 11 kW |
| $C_{r0}$ | 200 ηF | $L_{r0}$ | 20 μH |
| $L_{m0}$ | 120 μH | $n$ | 1.6 |
| $f_{max}$ | 200 kHz | $f_{min}$ | 60 kHz |
| $C_1$ | 100 μF | $C_{mosfet}$ | 0.75 ηF |

Based on the optimization algorithm, each of $L_r$, $C_r$, and $L_m$ has a global minimum for each operating point (each pair ($V_{bat}$,$P$)). Figures 13–15 show the global minimums of $L_r$, $C_r$, and $L_m$, respectively, according to the battery voltage and power variation.

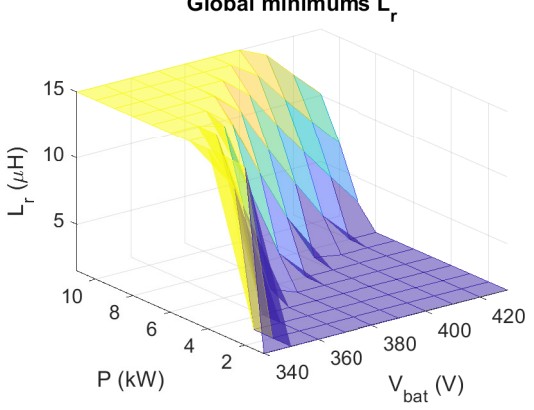

**Figure 13.** Global minimums of $L_r$ with respect to battery voltage and power variation.

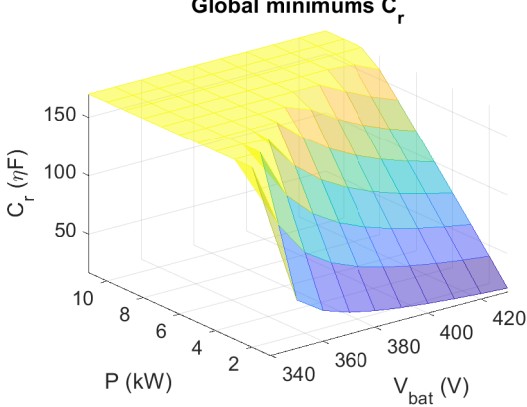

**Figure 14.** Global minimums of $C_r$ with respect to battery voltage and power variation.

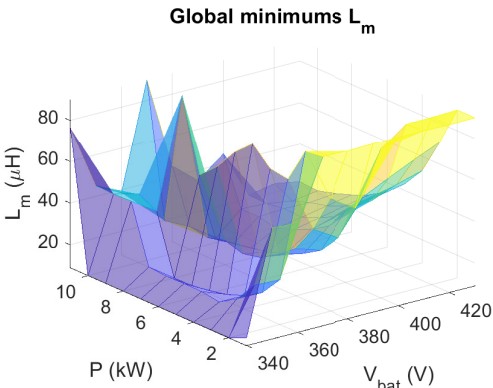

**Figure 15.** Global minimums of $L_m$ with respect to battery voltage and power variation.

The mean value of the global minimums for each variable, based on (37)–(39), can be obtained as follows:

$$L_{r1} = 6.668\,\mu\text{H}, C_{r1} = 76.62\,\eta\text{F}, L_{m1} = 35\,\mu\text{H}$$

It is confirmed that $L_r$ decreased by 67% (from $L_{r0}$ to $L_{r1}$), $C_r$ decreased by 62% (from $C_{r0}$ to $C_{r1}$) and $L_m$ decreased by 71% (from $L_{m0}$ to $L_{m1}$). Therefore, the cost and sizing of the EV charger decreased significantly.

With the mean values $L_{r1}$, $C_{r1}$ and $L_{m1}$, the feedforward switching frequencies $f_{0c}$ and $f_{0d}$ in G2V and V2X modes are calculated using Equations (15) and (24) according to the whole operating zone. $f_{0c}$ and $f_{0d}$ are presented in Figures 16 and 17, respectively, based on the optimized mean values $L_{r1}$, $C_{r1}$, and $L_{m1}$.

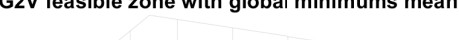
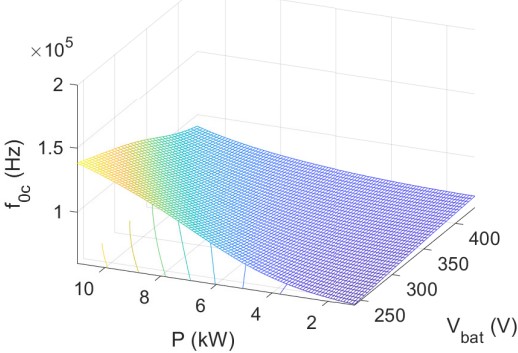

**Figure 16.** Feedforward frequency in G2V mode based on the optimized values.

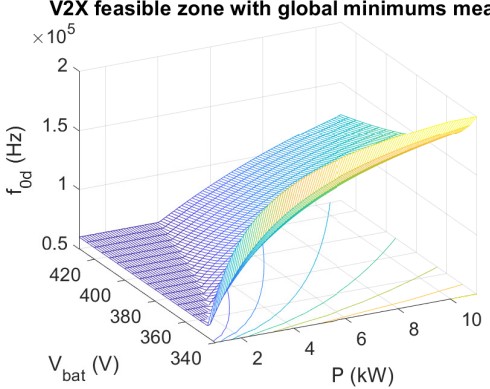

**Figure 17.** Feedforward frequency in V2X mode based on the optimized values.

It is clearly seen that, in Figure 16, $f_{0c}$ is completely within the feasible zone along the whole battery voltage and power variation when using the optimized values $L_{r1}$, $C_{r1}$, and $L_{m1}$, unlike in Figure 4 where there is a small saturation zone when using the initial values $L_{r0}$, $C_{r0}$, and $L_{m0}$. $f_{0c}$ in Figure 16 increases with the increase in the converter power for each battery voltage. On the other side, an important reduction in the saturation zone of $f_{0d}$ is achieved when using the optimized values as presented in Figure 17. There is a small saturation zone at $f_{min}$, while in Figure 6 there is an important saturation zone at $f_{max}$. Moreover, it is clearly seen that $f_{0d}$ is directly proportional to the converter power for each battery voltage, whereas in Figure 6 it is inversely proportional.

A comparative study of the DC-DC LLC converter efficiency between initial values ($L_{r0}$, $C_{r0}$, and $L_{m0}$) and optimized values ($L_{r1}$, $C_{r1}$, and $L_{m1}$) is completed according to the power variation for some battery voltages.

It should be noted that the AC-DC converter of the electric vehicle charger (Figure 1) is used to regulate the grid current, and consequently the DC bus current at the input of the DC-DC converter. The DC bus voltage is regulated by the DC-DC LLC converter to follow the setpoint (450 V). The modulation strategy, i.e., PFM strategy, is implemented based on the small signal modelling of the LLC converter to regulate the DC bus voltage by varying the switching frequency. The battery voltage is imposed by the Battery Management System (BMS). In G2V mode, the input power of the LLC converter is the product of the DC bus voltage and the DC bus current. The output power is the product of the battery voltage and the battery current. For each operating point, the efficiency is obtained by dividing the average value of the output power at the battery side by the average value of the input power at the DC bus side. Although in the V2X mode, the efficiency is calculated by dividing the average value of the output power at the DC bus side by the average value of the input power at the battery side.

Figures 18 and 19 show an efficiency comparison in G2V mode with $V_{bat}$ = 350 V and $V_{bat}$ = 390 V, respectively. It is confirmed that, at low powers, there is an important efficiency improvement when using the optimized values. At high powers, when P exceeds 5000 W in Figure 18 (with $V_{bat}$ = 350 V) and 8000 W in Figure 19 (with $V_{bat}$ = 390 V), the efficiency resulting from initial values is higher than that obtained from optimized values. Actually, looking back at Figure 16, it becomes clear that $f_{0c}$ resulting from optimized values increases in parallel with the converter power for each battery voltage. At low powers, it presents values lower than $f_{0c}$ resulting from initial values (Figure 4), whereas at high powers, it presents higher values for some battery voltage. Thus, although it is the case of optimized values, the switching frequency at high powers, with $V_{bat}$ = 350 V and $V_{bat}$ = 390 V, is higher than that in the case of initial values, giving higher switching losses and less efficiency as presented in Figures 18 and 19.

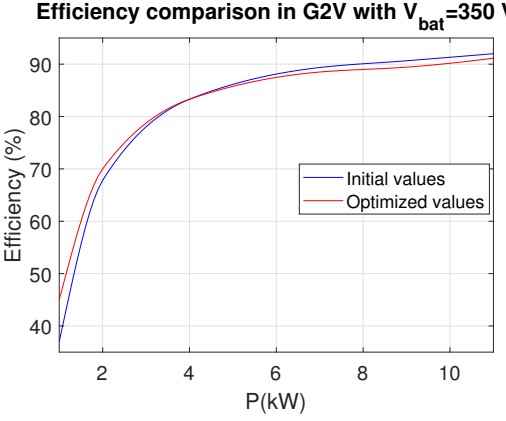

**Figure 18.** Efficiency comparison in G2V mode with $V_{bat}$ = 350 V.

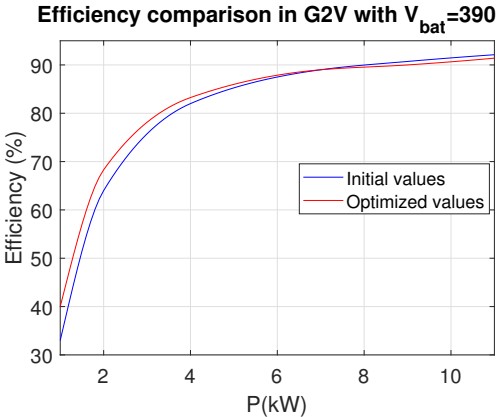

**Figure 19.** Efficiency comparison in G2V mode with $V_{bat}$ = 390 V.

Figures 20 and 21 illustrate an efficiency comparison in V2X mode. According to these figures, it is confirmed that, at low powers, there is an important efficiency improvement when using the optimized values. For P = 2000 W, the efficiency is 87.7% (with $V_{bat}$ = 350 V in Figure 20) and 86.3% (with $V_{bat}$ = 390 V in Figure 21), which is much better than the case of initial values where it is 78.78% (with $V_{bat}$ = 350 V) and 79.3% (with $V_{bat}$ = 390 V). When P is between 4000 and 8000 W, the efficiency resulting from optimized values is always higher than that resulting from initial values. It remains higher until P exceeds 9500 W for $V_{bat}$ = 350 V (Figure 20) and 10,500 W for $V_{bat}$ = 390 V (Figure 21), where it becomes a bit lower. It should be noted that $f_{0d}$ obtained using optimized values (Figure 17) increases with the increase in converter power, whereas it decreases when using initial values (Figure 6). Thus, when P exceeds 9500 W with $V_{bat}$ = 350 V (Figure 20) and 10,500 W with $V_{bat}$ = 390 V (Figure 21), the efficiency obtained from optimized values becomes lower than that obtained from initial values. This is due to the fact that the switching frequency becomes higher when using optimized values, giving higher switching losses in this operating zone.

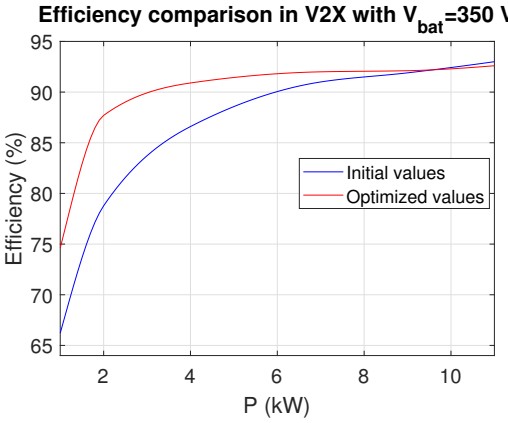

**Figure 20.** Efficiency comparison in V2X mode with $V_{bat}$ = 350 V.

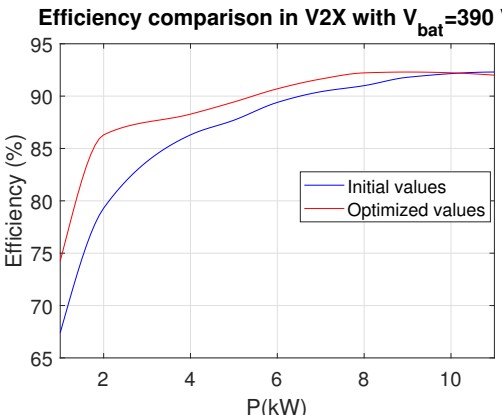

**Figure 21.** Efficiency comparison in V2X mode with $V_{bat}$ = 390 V.

## 5. Conclusions

In recent years, V2X technology for EV has been increasing in importance. A bidirectional charger structure, which includes a bidirectional DC-DC converter, is required to operate with high performances in both charging and discharging modes. The DC-DC LLC resonant converter is mostly adopted in industrial applications with high power density. An optimized design of the bidirectional DC-DC LLC resonant converter for EV charger application is proposed. The main goal is to reduce the charger cost and increase the soft-switching range for wide variation of the battery voltage and power. To our best knowledge, compared to what existing, this problem is not yet treated. It is an open problem coming from our industrial partner. The aim of this article is to solve it.

The small-signal modeling of the LLC converter with FHA is applied. PFM with gain inversion is adopted to regulate the DC bus voltage by varying the switching frequency. The decision variables are chosen considering the LLC resonant structure and the control frequencies in both G2V and V2X modes. The linear and non-linear constraints are defined based on the hardware and control requirements. An optimization methodology is generated for wide battery voltage and power ranges. The charger cost is minimized and the control frequency feasibility zones are improved. The optimization results show an important improvement in the charger efficiency and control performances in both G2V and V2X modes.

Perspectives for future work include the DC-DC LLC resonant converter design and control according to fixed switching frequency strategies such as Pulse Width Modulation PWM and Phase Shift Modulation PSM.

**Author Contributions:** Conceptualization, H.A.A.; Methodology, H.A.A., M.G. and M.T.; Software, H.A.A.; Writing—Original draft preparation, H.A.A.; Writing—Review and Editing, M.G. and M.A.H.; Visualization, M.G., M.A.H. and M.T.; data curation, H.A.A.; Supervision, M.G., M.A.H. and M.T.; Validation, M.G. and M.T. All authors have read and agreed to the published version of the manuscript.

**Funding:** This work was supported by the project Chair between Renault Group and Centrale Nantes about electric vehicle bidirectional charger control.

**Institutional Review Board Statement:** Not applicable.

**Informed Consent Statement:** Not applicable.

**Data Availability Statement:** The study did not report any data.

**Conflicts of Interest:** The authors declare no conflict of interest.

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
