# Peer review of "LLC DC-DC Converter Performances Improvement for Bidirectional Electric Vehicle Charger Application"

_wevj, doi:10.3390/wevj13010002_

Round 1
Reviewer 1 Report
In the paper, the method of optimizing components of resonant circuit determination is presented. Authors briefly described the state of art and then showed the basics of the method, which seems to be interesting and usable. Unfortunately, the paper is quite heavy to read and impossible to follow. The weak side of the paper is the lack of confirmation of the results in the laboratory test.
Few details that can improve the paper:
- there is no need to present formulas 4 and 14 in s domain, please keep ωt domain for analyzes,
- the formulas for foc from eqs 7-10 and fod from eqs. 17 and 18 should be added.
- the results of the sensitivity study in section 3.1 are interesting, in my opinion, the table with the shortly presented impact of the presented parameter should be added in the conclusion of this section
- the optimization method description should be developed in order to allow the reader to follow and use it.
- what are the criteria for choosing the initial values of parameters presented in Table 1
- the simulation should be described in more detail
- the efficiency plots (Fig 17-20) should be smooth
Author Response
Please see the attachement.

Reviewer 2 Report
Dear Authors,
Thank you for your good work. I have no objections, please follow the attachment and correct some typos in the text.
- In line 62 where you give the parameters, please refer to section 2 where is an explanation of its.
- mosfet - change to capital letters (MOSFET)
- figure 3 - delete the inscription next to the drawing

Author Response
Please see the attachement.

Reviewer 3 Report
Although the paper has improved, the following queries should be addressed.
1) Acronym V2H/L is not harmonized. Reshape the sentence as “to grid (V2G), to Home (V2H), to Load (V2L)”. The proper utilization of these terms (as long as V2X) has been discussed in doi: 10.3389/fenrg.2021.716389 and doi: 10.1016/j.enpol.2019.111136.
2) Is the battery voltage range (240-430V) actually compatible with commercial EV models?
3) From where did you take pictures? Make sure to use the same style and, in case, draw them again from scratch. Figure 2 presents voltage notation with the arrow pointing toward the negative point, while in Figures 3 and 5, the opposite notation is used. Capacitor and inductor symbol changes across figures, and voltage/current labels are not coherent.
4) Although the 3D plots look captivating, they are barely usable. Would you mind rotating plots to ensure optimal visibility of the 3D surface all the time? Moreover, you may want to add some contour plots embedded in the 3D plot or on the side (easily achievable using MATLAB or Octave) to make the plot readable. Use the same style.
5) Please provide a broader description of the efficiency plots. What assumptions did you implement? No description of the modeling of the switching device is given, and therefore evaluation of losses looks obscure. Is the efficiency plot somehow comparable with similar computation already available in the literature for the same topology? What is the efficiency wobbling that happens from 6 kW in Figure 21? Provide a broader description of Simulink settings to enable the reader to replicate what has been done. Consider introducing a screenshot (if pertinent and understandable) of the Simulink setup.
Round 2
Reviewer 1 Report
The paper is ready to publish
Author Response
We would like to thank you for the constructive remarks.
Reviewer 3 Report
The paper has improved, and it can go further. Just make sure you are citing EV DC charging standards like IEC 61851-23. In that standard (approved in 2014), the output voltage can reach 1000V and not only 400V.
